# Control-Talker: A Rapid-Customization Talking Head Generation Method for Multi-Condition Control and High-Texture Enhancement

## ABSTRACT

In recent years, the field of talking head generation has made significant strides. However, the need for substantial computational resources for model training, coupled with a scarcity of high-quality video data, poses challenges for the rapid customization of model to specific individual. Additionally, existing models usually only support single-modal control, lacking the ability to generate vivid facial expressions and controllable head poses based on multiple conditions such as audio, video, etc. These limitations restricts the models' widespread application. In this paper, we introduce a two-stage method called **Control-Talker** to achieve rapid customization of identity in talking head model and high-quality generation based on multimodal conditions. Specifically, we divide the training process into two stages: ***prior learning*** stage and ***identity rapid-customization*** stage. 1) In the prior learning stage, we leverage a diffusion-based model pre-trained on the high-quality image dataset to acquire a robust *controllable facial prior*. Meanwhile, we innovatively propose a high-frequency ControlNet structure to enhance the fidelity of the synthesized results. This structure adeptly extracts a high-frequency feature map from the source image, serving as a *facial texture prior*, thereby excellently preserving facial texture of the source image. 2) In the identity rapid-customization stage, the identity is fixed by fine-tuning the U-Net part of the diffusion model on merely several images of a specific individual. The entire fine-tuning process for identity customization can be completed within approximately ten minutes, thereby significantly reducing training costs. Further, we propose a unified driving method for both audio and video, utilizing FLAME-3DMM as an intermediary representation. This method equips the model with the ability to precisely control expressions, poses, and lighting under multi conditions, significantly broadening the application fields of the talking head model. Extensive experiments and visual results demonstrate that our method outperforms other state-of-the-art models. Additionally, our model demonstrates reduced training costs and lower data requirements.

## CCS CONCEPTS

• **Computing methodologies** → **Computer vision**.

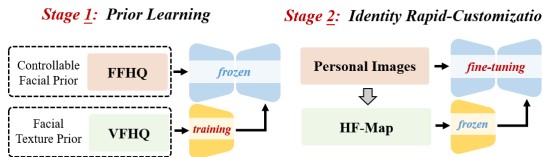

**Figure 1: Control-Talker involves two training stages: Prior Learning and Identity Rapid-Customization. The blue areas represent the U-Net part of the diffusion model, while the yellow areas denote the proposed HF-ControlNet. FFHQ and VFHQ are high quality image and video datasets.**

## KEYWORDS

Talking Head Generation, Multi-Condition Control, High-Frequency ControlNet, Rapid-Customization Diffusion

## 1 INTRODUCTION

Talking head generation is the task of animating a static source portrait to generate vivid expressions and controllable head poses guided by a driving video or audio segment. This is a challenging endeavor with critical implications across various domains such as video conferencing [6, 30, 37], virtual reality [7, 36], etc.

Recent works on talking head generation can roughly be classified into two categories: ***subject-agnostic*** and ***subject-dependent*** methods. 1) ***Subject-agnostic*** methods [24, 25, 27, 29, 33, 36, 39] are capable of driving any portrait by learning robust facial priors from large-scale video datasets. *The datasets required are often challenging to collect, and the computational resources needed for training are increasingly becoming difficult to afford, especially when training a talking head model capable of handling multi-condition inputs.* LipFormer [29] attempts to train on high-quality image datasets to reduce the substantial training costs. However, this approach constrains the synthesized area to the mouth and is limited to processing audio inputs only. StyleHEAT [33] proposes achieving adaptability to multiple conditions by performing inversion on a pre-trained StyleGAN [15] model. Nevertheless, errors in the inversion method often lead to modifications in identity and excessively smooth synthetic results. 2) ***Subject-dependent*** methods [22, 26, 28], on the other hand, focus on learning personalized head animation models by constructing datasets for a specific individual, ranging from several minutes to tens of hours in length. As shown in Table 1, while subject-dependent methods do not demand vast amounts of data, collecting high-quality videos of a specific individual is still time-consuming and labor-intensive. SynObama [28] collects approximately 16 hours of Obama's speech videos. Typically, it is not feasible to gather data of this scale for any other specific individual. Besides, both AD-NeRF [10] and DFRF [26] require several hours of training time, hindering their rapid transfer to other individuals.

**Table 1: Comparisons among several talking head generation model. Note that the Wav2Lip and SynObama can not generate talking head videos with only the driving video. "-" indicates that personalized training is not required or unknown. 'a' denotes subject-agnostic and 'd' signifies subject-dependent.**

| Method | Method Categories | Audio Driven | Video Driven | Exp&Pose Control | Light Control | Personalized Training Time | Personalized Dataset Size |
|---|---|---|---|---|---|---|---|
| Wav2Lip [24] | a | ✓ | | | | - | - |
| MakeItTalk [39] | a | ✓ | | | | - | - |
| FOMM [27] | a | | ✓ | ✓ | | - | - |
| StyleHEAT [33] | a | ✓ | ✓ | ✓ | | - | - |
| SynObama [28] | d | ✓ | | | | - | >16 h |
| AD-NeRF [10] | d | ✓ | | ✓ | | 36 h | >3 min |
| DFRF [26] | d | ✓ | | ✓ | | 7 h | >10 s |
| **Ours** | a & d | ✓ | ✓ | ✓ | ✓ | 10 min | 20 frames |

Moreover, due to the lack of a unified representation, these methods fail to generate videos with vivid expressions and poses driven by multiple conditions, i.e., video and audio.

In summary, both of the aforementioned methods are significantly limited by their excessive demands for high-quality video data and computational resources, and they generally fail to handle inputs under multi-condition scenarios (please refer to Table 1). Given a reconsideration of these issues, we propose a two-stage method called **Control-Talker** for the rapid customization and multi-condition control of talking head model. We innovatively propose that the training of the diffusion-based talking head generation model can be divided into *Prior Learning* and *Identity Rapid-Customization* stage, as shown in Figure 1.

1) *Prior Learning*. We decomposes the task of prior learning into two processes: *Controllable Facial Prior* learning and *Facial Texture Prior* learning. Firstly, we leverage a diffusion-based model pre-trained on a high-quality image dataset to learn the controllable facial prior. This learning process enables our model to possess the capability of implementing fundamental controls over expressions, poses, and lighting. However, the synthesized results often lack detailed textures, that is, over-smoothed phenomenon. Therefore, we secondly propose a well-designed high frequency ControlNet (**HF-ControlNet**) structure to learn the facial texture prior. This structure enhances the model's focus on high-frequency textures to enhance the fidelity of the synthesized results while preserving its controllability. Specifically, we segment the image to focus the model exclusively on the significant foreground portrait area. Subsequently, operations such as image sharpening and high-frequency filtering are employed to extract high-frequency texture feature map from the foreground portrait. This extracted feature map, along with physical buffers rendered by DECA decoder, are then fed into a structure similar to ControlNet. Through the steps above, HF-ControlNet facilitates the augmentation of high frequency textures in the synthesis results.

Leveraging the effective prior learning stage, our model facilitates controllable image generation with faithful texture preservation. To further enable our model to handle multi-condition inputs, we employ FLAME-3DMM as the intermediate representation, designing a unified driving method for multiple condition. This unified method comprises an easily implementable video-driven pipeline and an ingenious **Audio2FLAME** structure for audio-driven applications, which enhances the model's controllability and adapts it to a broader range of application scenarios.

2) *Identity Rapid-Customization*. Unlike other methods that gather extensive video data for training, we collect only 20 images of a specific individual from the internet or directly extract pictures of different poses from a video. Then through a few fine-tuning steps within merely 10 minutes, we fix the identity of the specific individual to achieve the goal of rapid customization. Furthermore, we retain the multi-condition control from the controllable facial prior and the texture enhancement from the facial texture prior, significantly enhancing the controllability and fidelity of the results.

Extensive experiments and ablation studies demonstrate that our method outperforms other state-of-the-art models in terms of training costs, synthesis quality, and identity preservation. The main contributions are summarized as follows:

- We propose a two-stage rapid customization method called **Control-Talker** for talking head generation.This method benefits from facial priors derived from high-quality image and video datasets, enabling rapid identity customization on a small subset of specific individuals.
- A high-frequency ControlNet architecture is introduced, which effectively preserves the texture details of the source image and enhances the fidelity of the results.
- By utilizing FLAME-3DMM as an intermediary representation, we design a unified driving method for multiple conditions, which enables the control of expression, pose, and lighting based on video and audio inputs.

## 2 RELATED WORK

### 2.1 Talking Head Generation

*Subject-agnostic methods*. Subject-agnostic methods aim to construct a universal methodology for driving arbitrary objects by learning robust facial structure and texture prior from large-scale datasets [7, 19, 24, 25, 36, 39]. However, it leads to an increasing demand for high-quality video dataset and unsustainable training costs, particularly when training a model capable of handling multi-condition inputs. Wav2Lip [24] introduces a reconstruction training strategy that synthesizing the talking head videos by inpainting the masked mouth area. However, artifacts are often present in the synthesis results, and there is no direct control over head pose and facial expressions. MakeItTalk [39] introduces an intermediary representation based on facial keypoints, which decomposes the task into two stages: mapping audio to keypoints and then synthesizing these keypoints into images. While MakeItTalk reduced the

**Figure 2: Overview of Control-Talker. Control-Talker utilizes FLAME-3DMM as the intermediate representation to achieve a unified video and audio driving approach, and can realize the editing of exp, pose and lighting. The U-Net part of the diffusion model is used to introduce a Controllable Facial Prior learned from high-quality image datasets, while the well-designed HF-ControlNet part incorporates facial texture priors by extracting high-frequency feature map from the source image.**

requirements for input driving video, it still failed to improve controllability or diminish the occurrence of artifacts. PIRenderer [25] proposes using 3D Morphable Model to control expressions and poses, achieving a unified driving method for both audio and video. However, this method requires frame-by-frame training on a large-scale video dataset, necessitating significant training costs. Unlike these methods above, we obtain controllable facial prior from a pre-trained diffusion model to reduce training costs and design a novel HF-ControlNet to learn texture facial prior for texture enhancement. Our method also uses FLAME-3DMM to unify multiple conditions, which provides expansion space for fine-grained control.

**Subject-dependent methods.** Unlike Subject-agnostic methods that generally focus on synthesizing arbitrary objects, Subject-agnostic methods emphasize the synthesis of specific individuals [10, 26, 28]. SynObama [28] demonstrates a noteworthy synthesis outcome by compiling 16 hours of speech videos of Obama. However, this technique's substantial data requirements hinder its applicability to other specific individuals. AD-NeRF [10] introduces a method based on 3D NeRF rendering, achieving the synthesis of a specific individual from a video spanning 3 to 5 minutes, reducing the difficulty of data collection. Nevertheless, the high training costs associated with AD-NeRF limit its broader adoption. Building upon the AD-NeRF framework, DFRF [26] incorporates a pre-trained NeRF model, enabling the fine-tuning of a specific individual's identity with approximately 10 seconds of short video, yet still necessitates around 7 hours of training. To reduce training costs and data requirements, our Control-Talker method acquires the robust facial priors from high-quality image and multi-view video datasets, enabling the identity rapid-customization stage to require only a mere 20 images and approximately 10 minutes of fine-tuning.

## 2.2 Diffusion Model on Face Image Generation

Denoising Diffusion Probabilistic Models (DDPM) [13] have made significant strides in the fields of image generation. Building upon

this, a series of methods for face image generation have been developed. DiffBir [21] utilize the powerful Stable Diffusion model as a prior, employing ControlNet to achieve facial image restoration. DiffusionRig [5] enhances the controllability of facial synthesis networks by integrating the FLAME model, and it improves the fidelity of synthesized images by utilizing the high-quality FFHQ image dataset for training. FaceX [11] enhances the Stable Diffusion model by incorporating an attribute decomposition network, enabling simultaneous processing of multiple facial editing tasks. Leveraging the powerful image synthesis capabilities of these diffusion model, we can significantly reduce the training costs and achieve enhanced high-frequency texture details through the proposed HF-ControlNet.

## 3 METHOD

In this section, we present our Control-Talker as shown in Figure 2. We first describe our design of the multiple conditions control, which encompasses a review of the 3D Morphable Face Models, along with introductions to both the Video-Driven and Audio-Driven (Audio2FLAME) pipelines (Section 3.1). Then, we provide a detailed introduction to the learning process of the facial texture prior from a well-designed high frequency ControlNet (HF-ControlNet) and the controllable facial prior from a pre-trained diffusion model(Section 3.2). Finally, we describe the process required for identity rapid customization (Section 3.3).

## 3.1 Multi-Condition Control

**3D Morphable Face Models.** 3D Morphable Face Models are widely used in talking head generation [25, 32, 38], image synthesis [4, 5, 20] and other fields, which can be used to accurately represent head posture, facial geometry, facial expression, etc. In this paper, we utilize a popular 3D Morphable Model (3DMM), FLAME [18], to precisely control aspects such as facial expressions, poses, and identities of the head. It takes shape $\beta$, pose $\theta$, and expression $\psi$ as inputs and outputs a full face mesh with $N$ vertices.

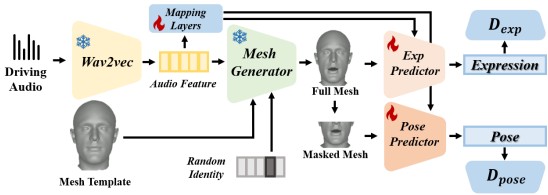

**Figure 3: The structure of the Audio2FLAME.**

The FLAME-3DMM model can be expressed as follows:

$$M(\beta, \theta, \psi) : \mathbb{R}^{|\beta| \times |\theta| \times |\psi|} \to \mathbb{R}^{3N} \quad (1)$$

Building on FLAME, the DECA model [9] further uses Lambertian reflectance and Spherical Harmonics (SH) lighting to represent the appearance of the face. As shown in Figure 2, we roughly divide the controllable parameters of the DECA model into identity(shape) $\beta$, expression $\psi$, pose $\theta$ and lighting $\lambda$. Therefore, through the DECA model, we can easily realize the basic control of face 3D model by changing pose, exp, lighting and other parameters.

**Video-Driven Pipeline.** Given a single source portrait $I_s$, and target video sequences $V_t = \{I_t^1, I_t^2, ..., I_t^N\}$, we first apply DECA to each frame of the source and target images to extract parameter sets $\{\beta, \psi, \theta, \lambda\}$. Subsequently, for any pair of source and target images, we can easily manipulate facial attributes by exchanging parameters $\{\beta_s, \psi_s, \theta_s, \lambda_s\}$ and $\{\beta_t, \psi_t, \theta_t, \lambda_t\}$ between them. For the purpose of talking head generation, $\beta s$ of the source image should be preserved to maintain the identity of source portrait. Then, the facial parameter set $\{\beta_s, \psi_t, \theta_t, \lambda_t\}$ is transmitted to the DECA decoder to render the physical buffers. These physical buffers contain identity information of the source image, as well as the expression, pose, and lighting details of the target image, thereby enabling precise control over attributes in subsequent image synthesis task. Other combined forms of the parameter set $\{\beta, \psi, \theta, \lambda\}$ can also be implemented in this way.

**Audio-Driven Pipeline.** We propose an **Audio2FLAME** model, as shown in Figure 3, to bridge the gap between audio and video. Unlike previous work, our model does not directly learn the mapping of audio to DECA expression $\psi$ and pose $\theta$ parameters, but leverages an off-the-shelf *Mesh Generator* to provide rich lip synchronization priors. To be specific, we first extract audio feature through the pre-trained Wav2vec [1] model. Then, a trainable linear layer maps the audio features to appropriate dimensions. Following this, the pre-trained Mesh Generator [8] receives the audio features, mesh template, and random identity as inputs and outputs the predicted complete head mesh. Here, the random identity represents the speaking style attributes.

We employ two structurally analogous 1D convolutional blocks to map the head mesh vertices to the expression and pose parameters, respectively. To decouple facial expressions from mouth movements, we retain only the lower half of the mesh points that are relevant to mouth motion for input into the pose predictor. In addition, the audio feature mapped by Mapping layers are further injected into the predictor through a cross-attention layer.

To mitigate potential prediction errors during the training process, we further employ audio sequences spanning a total duration of $2T$ frames, incorporating $T$ frames before and after the target frame, as input to the Mesh generator. This approach yields an output of head mesh vertices for the $2T$ frame interval. Subsequently, these $2T$ frame head mesh vertices are utilized to predict the expression and pose parameters of one target frame. Furthermore, we concurrently predict parameters for several successive frames, represented as $N$, to enhance temporal stability within a batch. Temporal discriminators $D_{exp}$ and $D_{pose}$ are also implemented follows the structure of PatchGAN [14].

The loss function for the Audio2FLAME model comprises two components: Reconstruction loss $L_{rec}$ and GAN loss $L_{gan}$. In the following formulas, pose $\theta$, and expression $\psi$ parameters are uniformly represented by the symbol $c$. $L_{rec}$ quantifies the error between the predicted $\hat{c}_t^{1:N}$ and ground truth $c_t^{1:N}$ parameters:

$$L_{rec} = \sum_{n=1}^{N} \|\hat{c}_t^n - c_t^n\|_2^2, \quad (2)$$

where $N$ is the length of consecutive frames in a training batch.

We use the temporal discriminator $D_{exp}$ and $D_{pose}$ to improve the fidelity and smoothness of the predicted parameters, which are trained jointly with the Audio2FLAME model:

$$L_{gan} = \mathbb{E}_{\hat{c}_t}[\log(1 - D(\hat{c}_t^{1:N}))]. \quad (3)$$

The total loss of Audio2FLAME model can be expressed as follows:

$$L_{all} = \lambda_{rec}L_{rec} + \lambda_{gan}L_{gan}, \quad (4)$$

where $\lambda_{rec} = 100$ and $\lambda_{gan} = 0.1$.

## 3.2 Prior Learning

In the prior learning stage, a pre-trained controllable image generation model is utilized to provide robust controllable facial prior and a well-designed high-frequency ControlNet (**HF-ControlNet**) is proposed to learn facial texture prior.

*3.2.1* **Controllable Facial Prior.** We use DiffusionRig [5] as the base image generation model, which take surface normals, albebo, and lamlertian render images as inputs, and then generate the coarse results. The surface normals, albedo, and Lambertian render images can be easily rendered using the DECA decoder through the aforementioned Video or Audio-Driven Pipeline. Although preliminary results can be obtained through the pre-trained model, the synthesized facial images lack detailed textures, that is, **over-smoothed phenomenon** (as shown in Figure 9 (a)). Consequently, it is necessary to further incorporate a learning process for facial texture priors to enhance the fidelity of talking head generation. The high-frequency ControlNet is designed to address this issue.

*3.2.2* **Facial Texture Prior.** We design the **HF-ControlNet** structure to introduce high-frequency texture details from the source portrait as facial texture prior. Given a source portrait $I_s$, we first employ the *Segmentor* to segment the foreground area $I_f$, effectively eliminating interference from the background area. Subsequently, we use Laplace kernel to obtain the sharpened image $I_{sharp}$, which includes enhanced detail texture. Then we utilize Sobel kernel to detect the edge information in the image following the method

**Figure 4: Multi-view dataset preprocessing pipeline.**

in [2], so as to obtain the high-frequency feature map $I_{hf}$. The whole process can be expressed by the following two formulas:

$$I_{sharp} = I_f + \lambda(I_f \otimes K_L), \tag{5}$$

$$I_{hf} = (I_{sharp} \otimes K_h + I_{sharp} \otimes K_v) \odot I_{sharp}, \tag{6}$$

where, $K_L$ denotes the Laplacian kernel, $\lambda$ represents the coefficient governing the sharpening effect. $K_h$, $K_v$ indicate the horizontal and vertical Sobel kernels, respectively, serving as high-pass filters. $\otimes$ ,$\odot$ refer to convolution and Hadamard product.

After obtaining the high-frequency feature map $I_{hf}$, we design a HF-ControlNet structure similar to the ControlNet network. This structure receives the high-frequency feature map $I_{hf}$ and physical buffers as inputs to introduce high-frequency texture information into the diffusion model. Please refer to the supplementary materials for detailed structure.

***Multi-View Dataset and Random Target Strategy.*** Instead of directly using the source high-frequency map in the HF-ControlNet training process, we propose to construct a multi-view dataset based on the existing video dataset to avoid the gap between image reconstruction and driving tasks (as shown in Figure 4). By sampling images from videos at equal intervals, we are able to obtain multi-view pictures of the same individual while simultaneously reducing the training costs for the model. When training the HF-ControlNet, we randomly select a target high-frequency feature map to serve as a texture supplement for the source image, achieving a mapping between the texture and the spatial positioning of the coarse physical buffers. The detailed process for data preparation can be found in the supplementary materials.

***Loss function.*** We utilize the reconstruction loss to train the proposed HF-ControlNet. Given a source image $I_s$, a random target high-frequency feature map $I_{hf}$, and source physical buffers denoted as $I_{pb}$, we can represent the model's condition set as $C = \{I_{hf}, I_{pb}\}$. The denoising network can be represented as $f_\theta$, which learns the reverse process of a Markov Chain of length $T$ and predicts noise at time $t : \epsilon_t$, where $t \in [1, ..., T]$. The predicted noise can be formally expressed as $\hat{\epsilon}_t = f_\theta(x_t, t, C)$, where $x_t = \alpha_t x_0 + \sqrt{1 - \alpha_t^2}\epsilon_t$, $\alpha_t$ is hyperparamter, $\epsilon_t$ is noise, $x_0$ represents the original image $I$ without any added noise. The training of HF-ControlNet is then modeled as a conditional denoising process optimized with the following objective:

$$L := E_{x, \epsilon \sim \mathcal{N}(0,1), C, t}[\|\varepsilon - f_\theta(x_t, t, C)\|_2^2]. \tag{7}$$

## 3.3 Identity Rapid-Customization

At this point, our Control-Talker is equipped with controllable facial prior and facial texture prior. The former is capable of generating controllable coarse facial images based on the input of physical buffers, while the latter enhances the texture details of the results through the high-frequency feature map extracted from the source image. To achieve the purpose of identity customization, we collect 20 images of an individual from the internet, featuring different poses and expressions, or by sampling 20 frames directly from a video. And then we fine-tune the U-Net part of the diffusion model, while keeping the parameters of the HF-ControlNet frozen.

When fine-tuning the diffusion model for identity customization, we also adhere to the *Random Target Strategy* to randomly select different high-frequency feature maps as supplements to the source images. We find that this strategy effectively prevents the model from overfitting to fixed poses.

## 4 EXPERIMENTS

### 4.1 Implementation Details

***Dataset.*** We select the datasets HDTF [38] and VFHQ [31] in our training and testing processes. We randomly select 10,000 videos from the VFHQ dataset to serve as the training data for the HF-ControlNet. For the training of the Audio2FLAME, we randomly choose 300 videos from the HDTF dataset. We extract the FLAME-3DMM coefficients for each frame of the dataset videos using the DECA model [9]. To train the AD-NeRF and DFRF methods, we collected several videos from YouTube, each with an average length of over 8 minutes. From these videos, we designate 3 minutes of videos as the training dataset for AD-NeRF, while a 10-second segment is selected as the training dataset for DFRF. We use another two 3-minutes Obama and Biden video clips which are not included in the training process as test set. Videos in training and testing processes are all processed using the facial alignment method of FFHQ [16] and then cropped to 256×256.

***Evaluation Metrics.*** We use the cosine similarity (CSIM) [36] of identity embdding between the source portrait and the generated results to evaluate identity preservation. Frechet Inception Distance (FID) [12], cumulative probability blur detection (CPBD) [23], and Learned Perceptual Image Patch Similarity(LPIPS) [35] are used to quantitatively measure the visual quality of the synthesized videos. The synchrony between audio and video is estimated by the SyncNet confidence [3]. Additionally, Average Pose Distance (AED) and Average Expression Distance (APD) is utilized to measure the facial motion accuracy [25].

***Training Details.*** We initialize the U-Net part of the diffusion model with DiffusionRig [5] model pretrained on the FFHQ dataset. We employ Adam optimizer [17] for all experiments. The learning rate for training Audio2FLAME is set at $1e^{-4}$, while the learning rate for HF-ControlNet is configured to $4e^{-5}$. In the identity rapid-customization stage, We fine-tune our model for 2,000 iterations with a batch size of 4. And the learning rate is decreased to $4e^{-6}$.

### 4.2 Comparison

In this section, we select the most frequently used individual, Obama, as the primary subject of the visualization, with Biden serving as

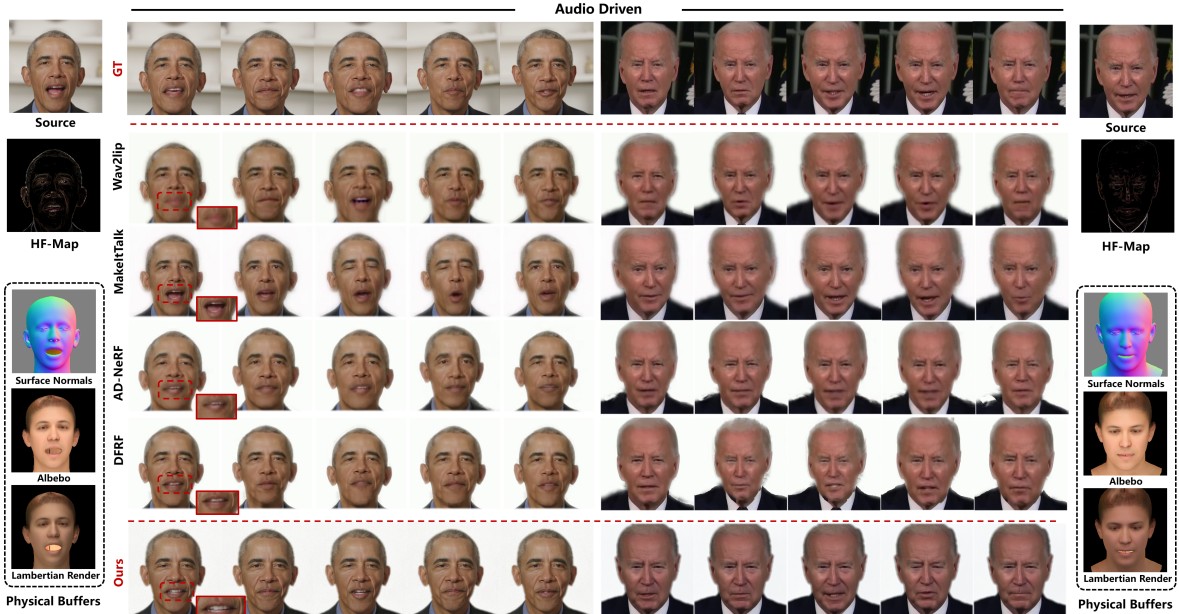

**Figure 5: Qualitative comparison with the state-of-the-art audio-driven methods. Our approach achieves optimal synthesis results in terms of image fidelity and identity consistency, and it also exhibits the highest similarity with the source images in terms of texture details, such as teeth and wrinkles (*please zoom in for a better view*).**

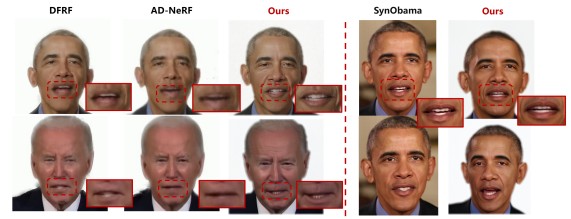

**Figure 6: Comparisons with the *Subject-dependent* methods in terms of detail texture.**

**Table 2: Quantitative comparisons with state-of-the-art audio-driven methods. Note that we evaluate Wav2Lip\* in the multi-driven setting, which means that Wav2Lip only synthesizes the mouth region. We utilize the 3-min Obama test set described in the Dataset section to conduct audio-driven talking head generation for the same identity.**

| Methods | FID↓ | CSIM↑ | LPIPS↓ | CPBD↑ | AED↓ | Sync↑ |
|---|---|---|---|---|---|---|
| Wav2Lip* [24] | **10.64** | **0.936** | **0.031** | 0.217 | 0.841 | **9.339** |
| MakeItTalk [39] | 25.68 | 0.792 | 0.289 | 0.202 | 1.704 | 3.132 |
| AD-NeRF [10] | 22.35 | 0.834 | 0.199 | 0.224 | 0.724 | 5.498 |
| DFRF [26] | 21.05 | 0.850 | 0.186 | 0.219 | 0.701 | 6.443 |
| **Ours** | 19.67 | 0.852 | 0.154 | **0.283** | **0.686** | 6.655 |

an additional supplement. The experiments are conducted in the context of audio-driven talking head generation, video-driven face editing and self-reconstruction.

*Audio-Driven Talking Head Generation.* We compare our method with several audio-driven methods, including Wav2Lip [24], MakeItTalk [39], AD-NeRF [10], and DFRF [26]. The first two are categorized as subject-agnostic methods, while the latter two are considered subject-dependent methods.

As shown in Figure 5, Wav2Lip [24] is capable of achieving high-degree lip synchronization. However, it often results synthetic artifacts around the mouth area, leading to unclear results. Similarly, while MakeItTalk [39] capable of predicting head poses through audio, it lacks explicit control mechanisms, leading to noticeable head jitter in the synthesized results. Our method, in contrast, produces more realistic results with the help of the effective facial texture prior learning. For the two NeRF-based methods, AD-NeRF [10] and DFRF [26], we observe that AD-NeRF is constrained by the clarity of the input video, preventing the synthesis of additional texture details. Conversely, the DFRF method exhibits artifacts around the head, as shown in Figure 5 Row 5, the synthesis results of Biden. Furthermore, we conduct a comparison with SynObama [28] and other subject-dependent methods in terms of detail texture (as shown in Figure 6). Due to the absence of available checkpoints, our comparison is directly with the best results showcased in SynObama's paper. Although SynObama undergoes training on the Obama dataset spanning 16 hours, the synthesized results still manifest blurriness like other methods, particularly in the area of the teeth (red area in the figure). In contrast, our model, owing to the prior learning stage on the high-quality datasets, is capable of synthesizing finer details.

We conduct quantitative comparisons on Obama test set in Table 2. Wav2Lip solely synthesizes the mouth region of the video,

**Video Driven: Self-Reconstruction**

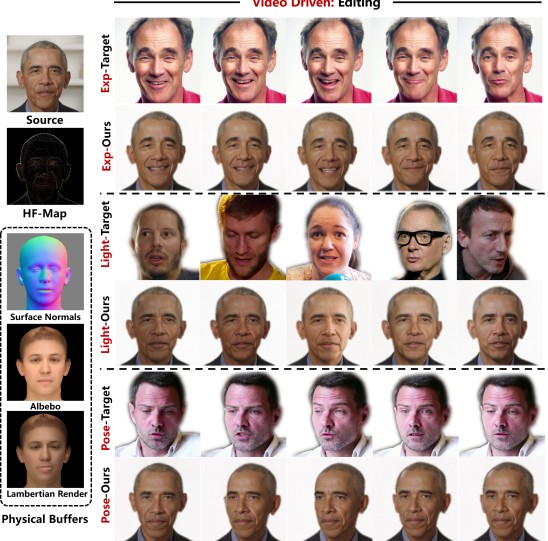

**Figure 7: Qualitative results of video-driven methods on the task of self-reconstruction.**

**Figure 8: Qualitative results of video-driven face editing. Left Column: the input source image, hf-map, and physical buffers. Row 1/3/5: the target video that provides the attributes of *expression, lighting,* and *pose,* respectively. Row 2/4/6: generated results.**

**Table 3: Quantitative comparisons with some representative video-driven methods on the self-reconstruction task.**

| Methods | FID↓ | CSIM↑ | LPIPS↓ | CPBD↑ | AED↓ | APD↓ |
|---|---|---|---|---|---|---|
| Bi-layer [34] | 99.53 | 0.426 | 0.615 | 0.077 | 1.203 | 0.045 |
| FOMM [27] | **27.13** | 0.847 | **0.118** | 0.109 | 0.785 | 0.044 |
| StyleHEAT [33] | 55.67 | 0.653 | 0.202 | 0.038 | 1.445 | 0.061 |
| **Ours** | 30.07 | **0.864** | 0.165 | **0.154** | **0.761** | **0.026** |

the Wav2Lip tends to generate artifacts in the facial area, particularly when performing inference at a resolution of 256x256, resulting in lower image clarity metric (CPBD). MakeItTalk synthesizes videos by predicting facial keypoints from audio, which leads to a decrease in identity consistency due to errors in keypoint prediction. Moreover, it lacks the capability for explicit control over posture and facial expressions. In contrast, our method utilizes the FLAME-3DMM as an intermediary representation, and the identity customization stage further ensures the preservation of identity, thereby resulting in an enhanced identity consistency metric (CSIM). Moreover, through the meticulously designed Audio2FLAME, our method circumvents the inaccuracies introduced by the prediction of discrete facial keypoints, thus enabling more precise predictions of expressions and poses. According to the results, AD-NeRF and DFRF tend to reconstruct expressions and poses already present in the training videos, which results in reduced generalizability. Our method, however, exhibits enhanced generalizability by pre-training on large-scale image and video datasets.

***Video-Driven Face Editing and Self-Reconstruction.*** We compare our method with some representative video-driven methods, including Bi-layer [34], FOMM [27] and StyleHEAT [33]. As shown in Figure 7 and Table 3, we can observe that although Bi-layer also employs a segmentation network to focus on synthesizing the foreground face, it fails to effectively preserve the identity from the source image, resulting in a CSIM of only 0.426. Additionally, there are significant artifacts at the junction between the foreground and background, see Figure 7 Row 2. FOMM can constrain the synthesized regions within the areas of motion through the predicted flow fields, thereby achieving higher reconstruction metrics. However, this method yields poor detail in the synthesized results, such as in teeth area (red areas in the figure). In contrast, our model is capable of preserving the texture from the original video by introducing a facial texture prior. StyleHEAT generates images by performing inversion on a pre-trained StyleGAN model. However, inaccuracies in the inversion method frequently result in alterations to the subject's identity and cause the final synthesized images to lack textural details. Our model, by contrast, preserves the high frequency texture through HF-ControlNet while retaining the identity well.

As our model incorporates both the controllable facial prior and facial texture prior, it is capable of editing expression and pose by implementing straightforward modifications to the FLAME-3DMM parameters. Furthermore, our method allows for varying lighting renders of the same video due to the introduction of lighting parameters in DECA, as demonstrated in the Figure 8 Row 4. This signifies that we can produce talking head videos with controllable expression, pose, and lighting, thereby broadening the scope of applications.

leaving the remainder identical to the target video. Consequently, it achieves the highest scores on the identity consistency metric (CSIM) as well as visual metrics such as FID and LPIPS. However,

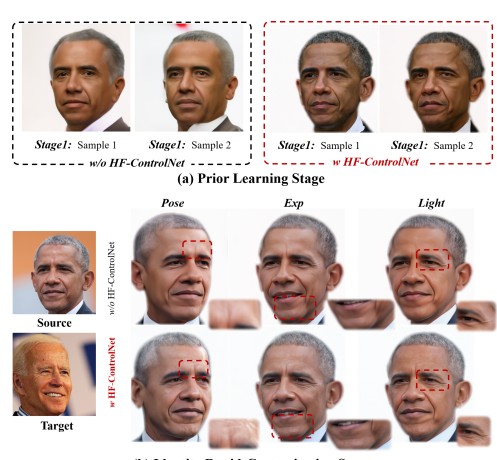

(a) Prior Learning Stage

(b) Identity Rapid-Customization Stage

Figure 9: Ablation study of the HF-ControlNet in the (a) *Prior Learning Stage* and (b) *Identity Rapid-Customization Stage*.

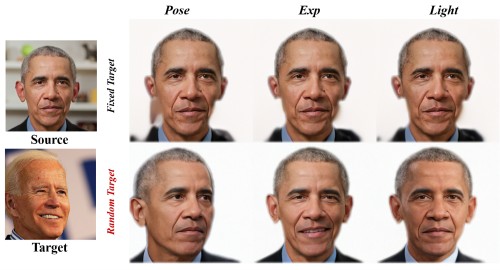

Figure 10: Ablation study of the Random Target Strategy.

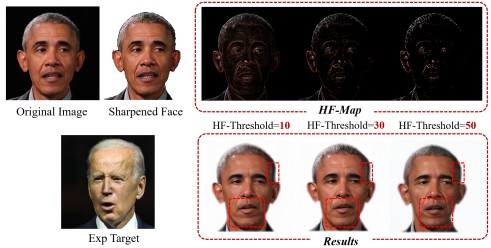

Figure 11: Visualization of the influence of different high frequency threshold.

## 4.3 Ablation study

We perform ablation studies to verify the effectiveness of several important designs in our method.

**Effect of HF-ControlNet.** In the prior learning stage, we randomly generate results without fixed identities for comparison. As shown in Figure 9 (a), results without utilizing HF-ControlNet (directly sampling from DiffusionRig) generally lack texture details, exhibiting over-smoothed surfaces. Our HF-ControlNet enhances the texture consistency by extracting high-frequency feature map

**Table 4: Ablation study on the effect of the HF-ControlNet and Random Target Strategy.**

| Methods | FID↓ | CPBD↑ | LPIPS↓ | APD↓ | AED↓ |
|---|---|---|---|---|---|
| Control-Talker | **32.12** | 0.303 | 0.186 | 0.029 | 0.810 |
| w/o Random Target | 33.36 | **0.440** | 0.372 | 0.137 | 1.650 |
| w/o HF-ControlNet | 38.43 | 0.221 | **0.141** | **0.027** | **0.648** |

**Table 5: Influence of different dataset sizes on Identity Rapid-Customization. We finally choose 20 as the size of the specific person dataset.**

| Dataset Size | FID↓ | CSIM↑ | CPBD↑ | LPIPS↓ | APD↓ | AED↓ |
|---|---|---|---|---|---|---|
| 5 frames | 34.42 | 0.830 | 0.250 | 0.239 | 0.034 | 0.837 |
| 10 frames | 34.49 | 0.831 | 0.259 | 0.190 | 0.031 | **0.738** |
| 20 frames | **32.12** | **0.835** | **0.303** | **0.186** | **0.029** | 0.810 |
| 40 frames | 32.25 | 0.824 | 0.262 | 0.214 | 0.036 | 0.765 |

from the source image, effectively preserving details such as wrinkles and teeth. After the identity rapid-customization stage shown in Figure 9 (b), our proposed HF-ControlNet still manages to capture more texture details from the Obama image, such as wrinkles and hairline, resulting in higher CPBD and FID scores. And our model demonstrates face editing capabilities consistent with the original DiffusionRig (w/o HF-ControlNet).

**Effect of Random Target Strategy.** We prevent the overfitting problem illustrated in Figure 10 by randomly selecting a target high-frequency feature map in a video during training. This strategy ensures textures can adapt to diverse poses, expressions, and lighting conditions, enhancing the model's controllability, as indicated by the AED and APD metrics in the Table 4.

**Influence of Different High Frequency Threshold.** We demonstrate the influence of high-frequency feature maps by setting various thresholds during the inference process. As shown in Figure 11, the content in the high-frequency feature maps correspondingly decreases with the increase of the high-frequency threshold. As a result, we ultimately selected a threshold of 10 to achieve better texture preservation.

## 5 CONCLUSION

In this paper, we propose a two-stage talking head generation method called **Control-Talker** for the identity rapid customization and multi-condition control. We leverage a diffusion model pre-trained on a high-quality image dataset to provide the controllable facial prior and design a high frequency ControlNet to learn the facial texture prior. Then, through a few fine-tuning steps, we achieve rapid customization of specific individual. Furthermore, we also design a unified driving method for multiple conditions, which enables the control of expression, pose, and lighting based on video and audio inputs. Experiments demonstrate the superiority of our method in terms of training costs, synthesis quality, and identity preservation.

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
