# OpenReview forum: "Control-Talker: A Rapid-Customization Talking Head Generation Method for Multi-Condition Control and High-Texture Enhancement"
_acmmm.org/ACMMM/2024/Conference — MM2024 Poster_

### Official Review · Reviewer_6oYm · 2024-05-04

**Rating:** 3
**Confidence:** 3

**Summary:**

This paper primarily introduces a method for talking face generation, the Control-Talker. This method allows for both one-shot generation and rapid customization for specific individuals (via fine-tuning a pretrained face generation model, the DiffusionRig). Besides, taking 3DMM as the intermediate representation, this method first predicts the 3DMM from the driving video or the driving audio and then renders the predicted 3DMM coefficients into the face images, supporting both audio-driven and video-driven talking face generation.

**Strengths:**

①This article is easy to understand.

②The proposed method is highly flexible, supporting audio-driven, video-driven, person-specific, and person-agnostic talking face generation.

③The proposed HF-ControlNet may further enhance the fine-grained texture of the generated faces.

**Limitations:**

### Weakness and Question
① This article appears to be an integration of several existing methods. For instance, it trains a predictor to predict 3DMM coefficients from audios or videos, and fine-tunes pretrained models (the DiffusionRig) to boost the performance on downstream tasks (i.e. using ControlNet to customize pretrained diffusion model). These techniques have been employed in numerous studies and seem somewhat old fashioned.

② The proposed HF-Controlnet does not align with its stated functionality to enhance fine-grained texture details.
The face images generated with and without the HF-Controlnet do not exhibit significant differences in texture detail, which may be due to the fact that these images are not of the same individual. To substantiate the effectiveness of the proposed module, more qualitative experimental results (with and without the proposed module with the same input source image $I_S$) should be provided.

③ The generated talking faces in the supplementary video exhibits noticeable jitter and poor interframe consistency. Additionally, it’s perplexing that the driving videos (the real videos) in the supplementary material also suffer from severe jitter. These jitters possibly comes from the face-centered cropping method.

④ Both quantitative and qualitative experimental results do not significantly beat baselines. Tables 2 and 3 indicate that the proposed method performs worse than one-shot methods (wav2lip) on some metrics. Furthermore, the proposed method only marginally outperforms existing person-specific methods. However, this is expected since the proposed approach heavily relies on pre-training priors (the pretrained DiffusionRig), while other baselines do not.

**Suitability:**

3

---

### Official Review · Reviewer_wHRN · 2024-05-18

**Rating:** 4
**Confidence:** 2

**Summary:**

The paper presents "Control-Talker," a novel approach for the generation of talking head models that allows for rapid customization and high-quality synthesis under multi-modal control conditions. The method is divided into two stages: the prior learning stage and the identity rapid-customization stage. In the prior learning stage, a diffusion-based model pre-trained on high-quality image datasets is used to obtain a robust facial prior, while a high-frequency ControlNet structure enhances the fidelity of the synthesized results by preserving facial texture. In the identity rapid-customization stage, the model is fine-tuned on a small number of images of a specific individual, completing the process within approximately ten minutes. Additionally, the paper introduces a unified driving method based on FLAME-3DMM, enabling precise control of expressions, poses, and lighting from both audio and video inputs. Extensive experiments show that the proposed method outperforms existing state-of-the-art models, reducing training costs and data requirements.

**Strengths:**

1. Multi-Modal Control Capability: Utilizing FLAME-3DMM as an intermediary representation for unified control of expression, pose, and lighting based on both audio and video inputs is a comprehensive solution that broadens the application fields of the model.

2. Rapid-Customization: The ability to customize the model's identity within ten minutes using only a few images is a substantial advancement, making the method more practical for real-world applications.

**Limitations:**

1. Lack of Discussion on Limitations: The paper lacks a thorough discussion on the limitations of the proposed method. Specifically, there is no analysis of the potential inefficiencies during the inference stage of the two-stage generation process. Understanding the trade-offs between training speed and inference efficiency is crucial for evaluating the practical applicability of the method in real-world scenarios. Including this discussion would provide a more balanced view of the method's performance and its possible constraints.

2. User Study for Realism and Perception: Incorporating a user study to assess the perceived realism and naturalness of the generated talking heads could provide valuable insights into the practical effectiveness of the method.

**Suitability:**

3

---

### Official Review · Reviewer_wrbU · 2024-05-20

**Rating:** 5
**Confidence:** 3

**Summary:**

This paper proposes a talking head generation method, called Control-Talker, which is able to support a wider variety of inputs than some previous generation methods. The paper describes the principle of each module in detail and shows rich visualization results. From the experimental results, the method can achieve certain results.

**Strengths:**

The object of this dissertation is the current popular method of talking head generation, which fully meets the requirements of multimedia and multimodal. The article is well organized, the description of the whole algorithm is more detailed, and the experiments are sufficient to prove the effectiveness of the design method.

**Limitations:**

While this article is finely organized as a whole, there are some details in it that the author omitted:
1) DECA is mentioned for the first time at L161, while the relevant citation is at around L363.
2) 'albedo' is misspelled several times in the paper, including but not limited to Fig. 2, at L444, Fig. 5, Fig. 8.
3) Equ. 1 omits '.' at the end.
4) The Comma at L487 appears at the beginning of the line.
5) AED and APD are incorrectly labeled at L566.
6) At L765 the letter x is used to indicate a multiplication sign.

Finally, some suggestions are provided to the authors for reference:

7) The black dashed box where Audio2FLAME points at Fig 2. can be modified to the same dark green dashed box as Audio2FLAME to indicate the source.
8) The comparison algorithm in Table 2 is not up-to-date. According to a recent measurement work (THQA: A Perceptual Quality Assessment Database for Talking Heads) it is stated that SadTalker (SadTalker: Learning Realistic 3D Motion Coefficients for Stylized Audio-Driven Single Image Talking Face Animation) is able to achieve better performance. Therefore I suggest the authors select at least 1 algorithm proposed in 2023 for comparison in this part of the experiment.

**Suitability:**

3

---

### Official Review · Reviewer_PZ7M · 2024-05-24

**Rating:** 3
**Confidence:** 2

**Summary:**

The paper proposes Control-Talker to achieve rapid-customization of identity in talking head model and high-quality talking head generation based on multimodal conditions. The high demand for computing resources in training talking head models, and the scarcity of high-quality video data, makes it a major challenge to quickly customize models for specific individuals. To address this issue, this paper proposes a novel two-stage strategy: 1) prior learning stage. Control-Talker extracts robust controllable facial prior from high-quality images through a diffusion-based model. The HF-ControlNet can learn facial texture priors, which helps to enhance high-frequency textures in the synthesized results; 2) Identity rapid customization stage. Control-Talker quickly achieves identity customization through a small amount of image data while maintaining multi condition control. In addition, Control-Talker utilizes FLAME-3DMM as an intermediate representation to achieve control of expressions, poses, and lighting in audio and video driven scenarios. The experimental results of the paper verified that Control Talker outperforms the compared models in terms of synthesis quality and identity preservation, and can complete model identity customization with minimal fine-tuning data and time. Control-Talker can promote the development of speech synthesis technology, pushing it towards more practical and wider applications.

**Strengths:**

This paper proposes Control-Talker, a two-stage method for generating talking heads, aimed at rapid-customization of identity and achieving multi condition control. By using a diffusion model pre trained on a high-quality image dataset to learn robust facial priors and introducing a high-frequency ControlNet structure to extract and enhance the high-frequency texture features of the source image, this method improves the fidelity and texture details of the generated results. Subsequently, with only a small amount of fine-tuning of specific individual images, identity customization can be quickly achieved, significantly reducing training costs and data requirements. The study also designed a multi condition driven method using FLAME-3DMM as an intermediate representation, enabling the model to control expressions, poses, and lighting under video and audio inputs, enhancing the practicality and application range of the model.

Compared to existing work, The main novelty of Control-Talker lies in its unique two-stage process, especially the proposal of high-frequency ControlNet and the implementation of multi condition unified driving using FLAME-3DMM, which are new developments in the field. Its contribution lies not only in technological breakthroughs, but also in opening up new directions for future research, such as how to more effectively integrate multiple conditional inputs to generate more natural dynamic faces, or how to further reduce the threshold for identity customization.

For ACM MM, this work closely aligns with the conference's focus on topics such as computer vision, multimodal interaction, and content generation. It demonstrates how to generate highly realistic talking head videos by integrating various data types such as videos and audio, which is in line with the topic of cross modal fusion, and has a direct contribution to promoting multimedia content generation technology.

**Limitations:**

The Control-Talker proposed in the paper demonstrates a novel and promising approach aimed at achieving rapid customization of head model identity and high-quality talking head generation based on multimodal conditions. However, regarding the writing of the paper, some content is not described clearly and appears to be somewhat confusing.

1)	According to line 23, "1) In the prior learning stage, we leverage a diffusion-based model pre-trained on the high-quality image dataset to acquire a robust controllable facial prior." This indicates that in the prior learning stage, the model needs to learn controllable facial priors. However, it is puzzling that in Figure 1, the part responsible for learning this controllable facial prior is frozen.

2)	When certain concepts that may not be well-known to the reader are first mentioned, they should be accompanied by citations (such as DECA, FFHQ, and VFHQ in the paper) or brief explanations (such as 'physical buffers' mentioned in the paper) to aid understanding.

3)	In line 266, there is an error in the expression of the sentence "Unlike Subject-agnostic methods that generally focus on synthesizing arbitrary objects, Subject-agnostic methods emphasize the synthesis of specific individuals". The "Subject diagnostic" is mistakenly repeated twice, and the second one should be corrected to "Subject dependent".

4)	In order to comprehensively evaluate the performance of the model, it is recommended to conduct more extensive audio driven experiments, such as using audio from different languages to drive the model (refer to GeneFace++), to explore the adaptability and generalization ability of the model.

5)	The experiment lacks direct comparison with the latest models, and the methods currently being compared are mainly based on technologies from 2022. Compared to current research progress, these methods appear relatively outdated.

**Suitability:**

3

---

### Meta-Review · Area_Chair_8N7K · 2024-07-03

**Recommendation:** Accept (Poster)
**Confidence:** 5

**Metareview:**

All reviewers agree that the proposed method is interesting and innovative. All reviewers comment that one-shot generation of talking head and rapid customization for specific individuals is highly flexible under multi-modal control conditions, which is novel and contribute significantly to the increased performance of the presented approach. The reviewers are satisfied with the presented experimental study. The rebuttal addressed a number of additionally raised questions. The authors are advised to include the relevant explanation in the final version of the paper. After reading the rebuttal, two of the reviewers upgraded the final score. Given a general appreciation of the work by the reviewers, I believe that the paper will be of interest to the audience attending ACM MM and would recommend a presentation of the work as a poster.